# Assessing readiness of health facilities for hypertension and integration with diabetes care in Bangladesh: Evidence from the National Service Provision Assessment Survey

Jannat-E-Mim Jisan[1], Mosiur Rahman[1]*, Md. Sarwar Zahan[2], Prosannajid Sarkar[3], Md. Rashed Alam[1], Wazedul Islam[1], Farhana Akhter Liza[1], Majharul Islam[1], Md. Mamunur Rashid[4], Anika Tabashsum[1], Chowdhury Mashrur Mahdee[1], Fahim Sahriar[5]

1 Department of Population Science and Human Resource Development, University of Rajshahi, Rajshahi, Bangladesh, 2 Institute of Educational Development, Brac University, Dhaka, Bangladesh, 3 Dr. Wazed Research and Training Institute, Begum Rokeya University in Rangpur, Rangpur, Bangladesh, 4 Department of Population Science, Jatiya Kabi Kazi Nazrul Islam University, Mymensingh, Bangladesh, 5 Department of Political Science, University of Rajshahi, Rajshahi, Bangladesh

* swaponru_2000@yahoo.com, mosiur@ru.ac.bd

## Abstract

### Background

Hypertension (HT) and diabetes mellitus (DM) are two major noncommunicable diseases (NCDs) with a high rate of comorbidity that utilize similar health system resources. Limited evidence exists, however, on the level of readiness of health facilities in Bangladesh to manage HT and integrated HT–DM care. The objective of this study was an assessment of facility readiness for HT and integrated HT–DM management and to identify key factors influencing levels of readiness.

### Methods

The study involved the analysis of data from 382 health facilities at or above subdistrict level from the most recent nationally representative dataset, the 2017 Bangladesh Health Facility Survey (BHFS). Readiness for HT and integrated HT–DM services was assessed based on composite scores as constructed from the WHO–Service Availability and Readiness Assessment (SARA) indicators. Negative binomial regression models were applied to determine the factors associated with facility readiness.

### Results

Facilities showed low readiness scores for both HT and integrated HT–DM care, with mean scores of 3.72 (out of 8) and 6.71 (out of 16), respectively. Although most facilities have basic equipment like BP apparatuses (98.7%) and stethoscopes (98.8%), a

**Data availability statement:** The dataset are publicly available from the DHS website: https://dhsprogram.com/data/available-datasets.cfm.

**Funding:** The author(s) received no specific funding for this work.

**Competing interests:** The authors have declared that no competing interests exist.

huge gap is observed in the training of staff (20.5%) and availability of guidelines for management (16.6%) of HT, diagnostic tools for DM, and essential medicines including ACE inhibitors (7.3%), thiazide diuretics (12.0%), and metformin (49.4%). Significant determinants of HT readiness included type of facility, client feedback system, and the number of HT care providers, while for integrated HT-DM readiness, important predictors were a type of facility, treatment-only service provision, client feedback mechanism, and structure of user-fees.

## Conclusions

In Bangladesh, health facilities are still not adequately ready to deliver integrated HT-DM services, illustrating deficits in human resources, clinical protocols, diagnostics, and availability of medicines at an overall system level. There are several areas that need improvement-the need for strengthening integrated training, free availability of medicines, client feedback systems, and collaboration between the public-private-NGO sectors. The results, though based on 2017 data, remain important in capturing system readiness prior to the reform and serve as a nationally representative baseline to assess the upspring of HT and NCD program improvements in Bangladesh.

## Introduction

Hypertension (HT), or elevated blood pressure, is recognized as one of the most eminent public health challenges globally and is the prime risk factor for cardiovascular diseases, stroke, kidney failure, and premature mortality [1]. It is estimated that about 1.28 billion adults between the ages of 30–79 now live with HT across the globe, most of them in low- and middle-income countries (LMICs) [2]. The growing burden of HT in LMICs, including countries like Bangladesh [3,4], continues to be spurred by urbanization, lifestyle changes, and population aging. Again, it has been proven that poor detection and control of HT would further worsen the existing load of noncommunicable diseases (NCDs) on health systems and countries' development [4]. Effective detection, treatment, and long-term follow-up of hypertensive patients are essential to reduce complications and improve health outcomes [5].

HT has been recognized as a major health concern in Bangladesh, but the management of the health system in this regard has remained less than optimal. A considerable proportion of HT individuals are either undiagnosed or under-treated, resulting in record low control rates and increased morbidity and mortality [4,6]. Previous studies have reported that many health facilities lack essential inputs for HT care—including trained personnel, treatment guidelines, diagnostic tools, and antihypertensive medicines—indicating low levels of readiness to provide adequate services [7–9]. Assessing facility readiness and identifying factors influencing service readiness are therefore crucial to strengthening the system's capacity for managing HT nationwide.

Although extensive research in Bangladesh has addressed the prevalence, awareness, and treatment of HT [4,10–16], most have focused on individual- or community-level determinants such as health literacy, treatment adherence, and healthcare access [15,16]. In contrast, nationally representative evidence on the readiness of health facilities to deliver HT services remains limited. Previous studies were mostly dealt with specific geographic areas [17] or specific types of facility [18] which usually restrict the generalizability to the national context. Bangladesh Health Facility Survey (BHFS) 2017 [19] is promising as it is based on the Service Provision Assessment (SPA) framework and therefore lends itself to adequate examination of the readiness of HT services under standardized domains such as staff training, equipment, diagnostics, and availability of essential medicines.

Also, diabetes mellitus (DM) and HT are two noncommunicable conditions that very often coexist in the same individuals [20]. Both share similar risk factors such as obesity, physical inactivity, and unhealthy diets and increase the risk considerably of developing cardiovascular disease and kidney failure as well as dying prematurely [21]. The coexistence of HT with DM includes needs for similar health system resources such as trained personnel, diagnostic tools, and essential medicines for a long-term management approach. Therefore, assessing DM care readiness of a facility alongside HT readiness is important for assessing the overall capacity of the health system to effectively manage chronic diseases.

However, previous analyses —both within Bangladesh and across countries—have typically examined HT and DM readiness separately [17,18,22–26]. Recognizing the strong epidemiological and service delivery linkages between these conditions, there is a growing global emphasis on integrating HT and DM services within health care. The World Health Organization offers a Package of Essential Noncommunicable (PEN) Interventions [27], and the Bangladesh National NCD Control Program has made such joint delivery of services between HT and DM a priority as a cost-effective strategy for improving NCD outcomes. Joint service delivery for HT and DM can enhance efficiency, optimum utilization of resources, and ensure continuity of care-particularly vital in resource-constrained health systems.

To address this critical evidence gap, the present study extends previous HT-focused research by assessing both (i) the readiness of health facilities to provide HT care and (ii) the degree of integration with DM services using nationally representative BHFS 2017 data. Furthermore, it identifies key determinants of both HT and integrated HT–DM readiness, by highlighting the extent of integration and the factors associated with service readiness, this study provides new insights for policymakers to advance integrated NCD care and strengthen Bangladesh's progress toward Universal Health Coverage (UHC).

## Methods

### Study design

This study used a cross-sectional design based on health facility data.

### Data source

This study used data from the BFHS 2017, which remains the most recent nationally representative survey [19] that includes standardized indicators of HT and DM service readiness. No subsequent nationally representative facility-level dataset with comparable measure has been released, making BHFS 2017 the only possible alternative for such analysis. Although these data reflect conditions that existed several years previously, they nevertheless represent important evidence for establishing a baseline for pre-reform system capacity. This creates a reference in comparison with current health system reforms such as the establishment of NCD corners, the scale-up of the WHO-PEN protocol, and integration of HT and DM services that began after 2017. Details on how to access and use of SPA data are available at https://dhsprogramcom/data/Using-DataSets-for-Analysiscfm. The BFHS 2017, aimed at generating detailed data on availability and readiness of health facilities in relation to tuberculosis, some non-communicable diseases (including HT), family planning, and maternal and child health in Bangladesh [19].

The survey collected information on staff availability and basic infrastructure, logistics (including supplies and essential medicines), laboratory services, and infection control protocols according to standard operating procedures of healthcare facilities. Key instruments for data collection in BFHS 2017 included facility inventory questionnaires, healthcare provider interviews, observation procedures, and exit interviews. Data from the facility inventory questionnaire was used for this study. Service readiness defined as the availability of laboratory tests, medicines, equipment, and functional services was evaluated by trained interviewers. Information was gathered through interviews with the facility manager, the person in charge, or the senior health professional responsible for client services at each facility.

The BFHS 2017 identified 19,811 registered facilities across the seven administrative divisions of Bangladesh: Barisal, Chittagong, Dhaka, Khulna, Rajshahi, Rangpur, and Sylhet [19]. The BHFS 2017 covered an extensive range of national facility types, including district hospitals, upazila health complexes, maternal and child welfare centers, facilities run by NGOs, private hospitals, and so on. However, specialized centers such as diabetic hospitals were not classified as a separate facility type. But under the BHFS sampling framework they included diabetes clinics and hospitals under the general categories of private facilities. Therefore, while these specialized centers were represented in the dataset, their information was aggregated within general facility groups rather than analyzed separately, which may limit the ability to distinguish their specific contribution to NCD service readiness.

The BFHS 2017 used a stratified random sampling technique, with stratification by organizational unit and facility type, to select 1,600 facilities from 19,184 eligible facilities in the formal sector. For this specific analysis on HT and integrated HT-DM readiness, the sample was refined. Since NCD services, including HT and DM management, are usually provided at the sub-district level and above in Bangladesh [19], facilities below the sub-district level (such as union subcenters, rural dispensaries, and community clinics) were excluded. Furthermore, facilities with missing data on key HT and integrated HT-DM readiness indicators were also excluded. After applying these criteria, the final sample was made up of 382 health care sites for analysis (**Fig 1**).

## Measures

**Outcome variables.** The outcome variables for this study were the readiness of health facilities to manage HT and integrated HT–DM care. The readiness was operationalized by composite measures based on essential indicators for provision of HT and DM care, as assessed through the BHFS 2017. These indicators were organized into three domains for HT and four for DM following the WHO-SARA framework [28], which is used to assess health facility service availability and readiness.

The staff and guidelines are the first domains for HT readiness and entail the presence of at least one health care worker trained in HT diagnosis or treatment within the last 24 months and the existence of HT-specific guidelines for diagnosis or treatment. The other domain is equipment and supplies, where the assessment checks the presence of a BP monitoring device (functional, whether digital or manual), an adult weighing scale, or a stethoscope. The third domain is medicines and commodities, which includes assessing the availability of at least one of the following: an angiotensin-converting enzyme (ACE) inhibitor, a thiazide diuretic, or a calcium-channel blocker (amlodipine or nifedipine).

DM readiness was assessed along four domains: staff and guidelines domain, which is similar to HT, capturing the availability of trained staff for DM care and the presence of DM-specific treatment guidelines; the equipment and supplies domain included availability of key DM care tools, such as BP apparatus, adult weighting scale, and adult height scale; the diagnostic tools domain assessed availability of essential DM diagnostic tools like glucose test strips, urine protein test, and urine glucose test; and finally the medicines and commodities domain examined availability of critical DM medications, including metformin, glibenclamide, injectable insulin, and injectable glucose solutions.

All the indicators were expressed as binary variables such that they could be present or not during the interviewer assessment with the facility records. The readiness score for each facility was obtained by summing the various indicators present in each domain. The final HT readiness score could range from 0 to 8 (covering the three HT domains). For

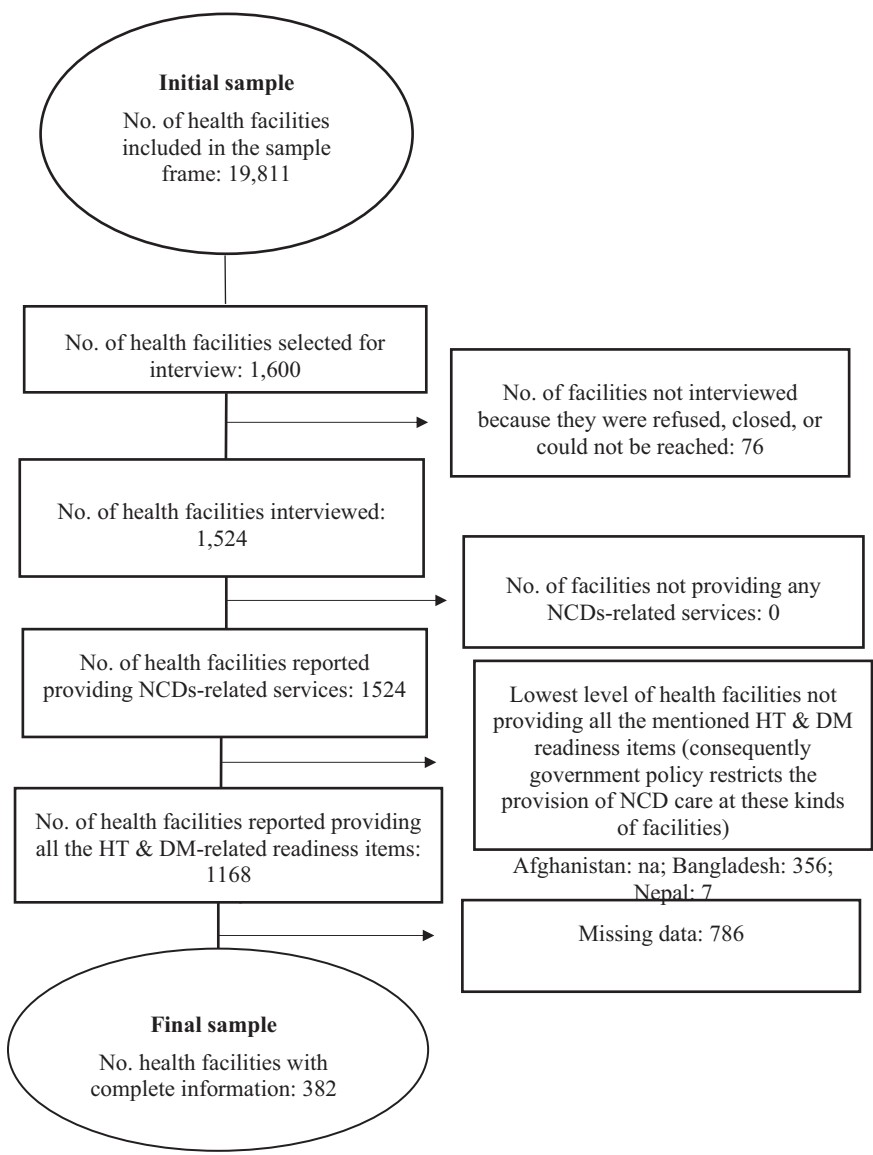

**Fig 1. Selection of the sample for HT and integrated HT-DM readiness.**

HT-DM readiness, the present study combined the indicators in both the HT and DM domains, enabling assessment of the overall facility readiness for co-managing both conditions. This integrated measure included staff and guidelines, equipment and supplies, medicines and commodities, and diagnostic tools domains for both HT and DM. The total integrated HT-DM readiness score ranges from 0 to 16, which reflects the cumulative availability of all indicators required for managing both conditions together.

**Explanatory variables.** Several characteristics were considered as possible explanatory factors that potentially influence HT and integrated HT-DM service readiness. All variables were obtained from the BFHS 2017 dataset [19] and selected based on prior research on facility readiness for chronic disease care [17,22–26]. Facility location was categorized as urban or rural and managing authority distinguished between public (government-managed) or private (non-governmental). External sources

of revenue categorized whether facilities received additional financial assistance from non-governmental donor agencies, the government, or neither. Quality assurance activities and external supervision were each measured as binary variables, denoting whether the facility routinely conducted internal reviews (e.g., mortality audits, record checks) and whether it had been supervised by higher health authorities within the previous six months, respectively.

User-fees structure was grouped into three categories: no fee, single fee per service, and fixed fee for all services. grouped into three categories. Availability of health personnel 24 hours a day was treated as a binary variable (yes/no), indicating whether an accredited health professional was on duty or on call for emergencies. Client feedback system was also coded as a binary, indicating whether the facility collected and reviewed patient opinions. The diagnostic and treatment capability variable captured whether facilities could diagnose only, treat only, or both diagnose and treat HT. For the integrated model, a corresponding measure of integrated HT-DM service capacity was used distinguished facilities that provided diagnosis only, treatment only, both diagnosis and treatment, or mixed/partial services for both conditions. Facility type was categorized as NGO clinics/hospitals, private clinics/hospitals, upazila health complex, and district hospitals. Finaly, the number of HT care providers and the total number of HT and DM care providers were included as continuous variable.

## Statistical analysis

First, descriptive statistics were used to summarize the sample, that is, means and standard deviations for continuous variables and weighted proportions for categorical variables. Two negative binomial regression models were built to determine factors associated with HT readiness and integrated HT-DM readiness.

The dependent variables were scores for HT readiness (0–8) and integrated HT-DM readiness (0–16). Preliminary tests for each score indicated over-dispersion (variance > mean), making Poisson regression inappropriate. The negative binomial model was adopted because its dispersion parameter (α) accounts for extra-Poisson variation, and the parameter was statistically significant in both models [29]. Model fit statistics confirmed the superiority of the negative binomial model compared to the Poisson model, as indicated by lower Akaike Information Criterion (AIC) values. Results are given as incidence rate ratios (IRRs) with 95% confidence intervals (CIs). The IRR shows the multiplicative change in expected readiness score for a one-unit increase in the value of a continuous predictor or for the presence of a specific category in a categorical predictor, while controlling for the other variables.

All independent variables were entered simultaneously into multivariable models to estimate independent associations. A $p$-value < 0.05 was considered statistically significant. To obtain national estimates representative across facility types and geographic areas, facility-level sample weights were applied to account for the stratified design of the BHFS 2017 study. All statistical analyses were performed using Stata version 16.0 (StataCorp, College Station, TX, US).

## Ethical considerations

The study made use of secondary, anonymized data from the BFHS 2017, which is a publicly available data set. Since this data was devoid of identifying characteristics of any individual either from the facility staff or from the patients, ethical approval from a research ethics committee was not required for the analysis. The BFHS 2017 survey was approved by USAID, Macro International, and the institutional review boards of the respective ministries of health in Bangladesh. The data collection procedures conformed to the ethical principles formulated in the 2013 updates to the Declaration of Helsinki. Informed consent was obtained from all participants (facility managers or designated staff) ahead of their involvement in the survey.

## Results

### Descriptive statistics

Table 1 provides an overview of the background characteristics of 382 surveyed health facilities in Bangladesh. Most facilities (66.5%) were in urban areas and 64% were privately managed. NGO clinics/hospitals accounted for 43.3%, private clinics/hospitals 27.2%, upazila health complexes 25.6%, and district hospitals 3.9%. About 71.8% of facilities received

**Table 1. Percentage distribution of surveyed facilities according to background characteristics: BHFS 2017 (n = 382).**

| Facility characteristics | n | % |
|---|---|---|
| **Facility location** | | |
| Rural | 117 | 33.5 |
| Urban | 265 | 66.5 |
| **Managing authority** | | |
| Private | 169 | 64.0 |
| Public | 213 | 36.0 |
| **Facility type** | | |
| NGO clinic/hospital | 100 | 43.3 |
| Private clinic/ hospital | 83 | 27.2 |
| Upazila Health Complex | 137 | 25.6 |
| District Hospital | 62 | 3.9 |
| **External sources of revenue** | | |
| None | 7 | 3.3 |
| Other than Govt. | 217 | 71.8 |
| Govt. | 158 | 24.9 |
| **Client fees** | | |
| No fees | 39 | 7.4 |
| Separate fees | 307 | 85.9 |
| Fixed for all services | 36 | 6.7 |
| **External supervision (past 6 months)** | 343 | 83.5 |
| **Routine quality-assurance activities** | 176 | 41.7 |
| **24-hour provider availability** | 283 | 61.2 |
| **Client's opinion reviewed** | 61 | 16.2 |
| **Ability to provide HT care** | | |
| Diagnosis only | 44 | 17.4 |
| Treatment only | 6 | 1.3 |
| Both diagnosis and treatment | 332 | 81.3 |
| **Ability to provide DM care** | | |
| Diagnosis only | 59 | 19.6 |
| Treatment only | 10 | 1.1 |
| Both diagnosis and treatment | 313 | 79.3 |
| **Ability to provide integrated HT-DM care [1]** | | |
| Diagnosis only for both conditions | 34 | 14.0 |
| Treatment only for both conditions | 3 | 0.5 |
| Both diagnosis and treatment for both conditions | 300 | 75.1 |
| Mixed/partial | 45 | 10.4 |
| **Number of HT care providers, Mean (± SD)** | 3.51 (±3.55) | |
| **Number of DM care providers, Mean (± SD)** | 3.20 (±2.52) | |
| **Total NCD care providers (HT & DM), Mean (± SD)** | 6.71 (±5.66) | |

[1]Integrated service capacity combines the two disease-specific variables into categories: 1 = diagnosis only, 2 = treatment only, 3 = both diagnosis and treatment, 4 = mixed/ partial. Percentages are weighed to national facility distribution; counts are unweighted.

non-governmental funding. The majority (85.9%) charged separate fees, while 83.5% received external supervision and 41.7% conducted routine quality assurance. Only 61.2% had 24-hour provider availability and 16.2% reviewed client feedback. As for the healthcare services, 81.3% of the facilities were capable of diagnosing and treating HT. Similar patterns were observed with diabetic management, where 79.3% of the facilities provided diagnosis and treatment. In addition, integrated HT-DM care can be provided by 75.1% of the facilities, with 10.4% providing mixed or partial service. The mean number of HT cares per facility was 3.51 (SD = 3.55), whereas the mean DM care specialists were 3.20 (SD = 2.52). Overall, the average pooled total NCD care providers for both HT and DM per facility was 6.71 (SD = 5.66).

Table 2 indicates the availability of key components for HT, DM management, and integrated HT-DM care readiness among health facilities in Bangladesh. In the staff and guidelines category, only 20.5% of facilities had at least one trained staff member for HT care, and 16.6% had treatment guidelines concerning HT management. For integrated HT-DM care, the availability of trained staff or guidelines fell to 10.1% or 16.1% of facilities. For equipment and supplies, 98.7% of the facilities had functional BP apparatus, 94.3% adult weighing scales, and 98.8% stethoscopes. However, adult height

**Table 2. Availability of HT, DM, and integrated HT–DM readiness indicators among health facilities, Bangladesh BHFS 2017 (n = 382).**

| Readiness domains & indicators | HT readiness (%) | DM readiness (%) | Integrated HT–DM readiness (%) |
|---|---|---|---|
| **Staff & guidelines** | | | |
| At least one trained staff for HT/DM care | 20.5 | 20.5 | 10.1[1] |
| Availability of treatment guideline for HT/DM | 16.6 | 23.9 | 16.1[2] |
| *Mean domain score* | 0.37 ± 50 | 0.44 ± 59 | 0.81 ± 0.97 |
| **Equipment & supplies** | | | |
| Functional BP apparatus | 98.7 | 98.7 | 98.7 [3] |
| Adult weighing scale | 94.3 | 94.3 | 94.3 |
| Stethoscope | 98.8 | -- | 98.8 |
| Adult height scale | -- | 69.5 | 69.5 |
| *Mean domain score* | 2.92 ± 0.31 | 2.63 ± 0.59 | 3.61 ± 0.62 |
| **Diagnostic tools** | | | |
| Urine Protein test | -- | 14.1 | 14.1 |
| Urine Glucose test | -- | 11.4 | 11.4 |
| Glucose test strips available | -- | 59.1 | 59.1 |
| *Mean domain score* | -- | 0.85 ± 0.78 | 0.85 ± 0.78 |
| **Medicines & commodities** | | | |
| ACE inhibitor | 7.3 | -- | 7.3 |
| Thiazide diuretic | 12.0 | -- | 12.0 |
| Amlodipine/Nifedipine | 24.6 | -- | 24.6 |
| Metformin | -- | 49.4 | 49.4 |
| Glibenclamide | -- | 11.8 | 11.8 |
| Injectable Insulin | -- | 13.6 | 13.6 |
| Injectable glucose solution | -- | 24.9 | 24.9 |
| *Mean domain score* | 0.44 ± 0.82 | 0.99 ± 1.14 | 1.44 ± 1.82 |
| **Overall readiness score** | 3.72 ± 0.94[4] | 4.91 ± 1.61[5] | 6.71 ± 2.18[6] |

[1, 2]If both HT and DM staff or guidelines are available, the answer is **"Yes"** for **integrated readiness.** If either of the two is not available, the answer is **"No"**; [3]Since the item is the same for both HT and DM, it is counted **only once** in the **integrated readiness**; [4]Scores are calculated per facility on a 0–8 scale for HT readiness; [5]Scores are calculated per facility on a 0–11 scale for DM readiness; [6]Scores are calculated per facility on a 0–16 scale for integrated readiness. Percentages are weighted to national distribution; counts are unweighted.

scales were available in only 69.5% of the facilities. Only 14.1% of the facilities had a urine protein test, 11.4% offered urine glucose tests, and 59.1% had glucose test strips.

With respect to medicines and commodities, only 7.3% of facilities had ACE inhibitors, 12% had thiazide diuretics, and only 24.6% had calcium channel blockers (nifedipine) in the facility. Further, only 49.4% of facilities had metformin, and 13.6% provided injectable insulin. Overall, the readiness score was 3.72 out of 8 in HT care. The average readiness scores for DM care (4.91/11) and integrated HT-DM care (6.71/16) show that there is a similar suppressive trend in the different domains. HT readiness score stands at 46.5% ($3.72 \div 8 \approx 46.5\%$) of the maximum possible score, and the integrated HT–DM readiness score is at 41.9%, ($6.71 \div 16 \approx 41.9\%$) indicating similar trends of low readiness in both types of care.

### Multivariate analysis

Table 3 presents the results of a negative binomial regression analysis examining factors associated with health facility readiness to manage HT and integrated HT-DM care.

**For HT readiness.** NGO clinics/hospitals (IRR = 0.83; 95% CI = 0.74–0.92), upazila health complexes (IRR = 0.76; 95% CI = 0.64–0.92) and district hospitals (IRR = 0.79; 95% CI = 0.66–0.94) demonstrated significantly lower readiness compared to private clinics/hospitals. Facilities with a client feedback system (IRR = 1.12; 95%CI = 1.03–1.21) were more likely to be ready for HT care, and a higher number of HT care providers was also significantly associated with greater readiness (IRR = 1.01; 95% CI = 1.00–1.02).

**For integrated HT-DM readiness.** NGO clinics/hospitals (IRR = 0.82; 95% CI = 0.64–0.98) showed lower integrated readiness compared to private clinics/hospitals. Facilities that offered treatment-only for both conditions (IRR = 1.28; 95% CI = 1.04–1.52) exhibited higher integrated HT-DM readiness. Additionally, facilities that reviewed client feedback (IRR = 1.17; 95% CI = 1.04–1.30) and charged separate client fees (IRR = 1.14; 95% CI = 1.01–1.28) demonstrated greater integrated readiness.

## Discussion

### Main findings

This is a nationally representative survey assessing the readiness of health services to provide HT and integrated HT-DM care in Bangladesh, underlining persistent gaps in some of the essential service components. Basic equipment such as BP apparatuses and stethoscopes was readily available; however, there were major shortages of trained personnel, clinical guidelines, and essential medicines for both HT and DM, as well as diagnostic tools specifically for DM. On the average readiness score, facilities generally remain poorly ready for both autonomous HT and integrated HT-DM care. Multivariable analyses showed that facility type, client feedback system, and number of HT care providers were the significant determinants of HT readiness. Facility type, facilities offering treatment-only services for both conditions, those collecting client feedback, and those who charged separate service fees demonstrated relatively higher integrated HT-DM care readiness.

### Novelty and contributions

This study is among the first in Bangladesh to assess not only HT service readiness but also the integration of HT and DM care using nationally representative data. Earlier studies had analyzed these conditions separately; however, the current study employs a WHO-SARA-based composite index that assesses both conditions, thus providing an overall picture of health system readiness for both NCDs management.

Unlike previous regional studies that compared Bangladesh with other South Asian settings using the same BHFS dataset to assess health facilities readiness for DM and cardiovascular diseases, this study provides a Bangladesh-specific view, enabling a more detailed examination of system gaps such as the lack of national treatment guidelines and

**Table 3. Determinants of HT and integrated HT–DM readiness among health facilities, Bangladesh BHFS 2017 (n = 382).**

| Variables | IRR (95% CI) – Hypertension readiness¹ | p-value | IRR (95% CI) – Integrated HT–DM readiness² | p-value |
|---|---|---|---|---|
| **Facility location** | | | | |
| Rural | 1.00 | | 1.00 | |
| Urban | 1.00 (0.94-1.07) | 0.917 | 1.04 (0.95-1.13) | 0.415 |
| **Managing authority** | | | | |
| Private | 1.00 | | 1.00 | |
| Public | 0.97 (0.84-1.10) | 0.448 | 0.92 (0.80-1.05) | 0.218 |
| **Facility type** | | | | |
| Private clinic/hospital | 1.00 | | 1.00 | |
| NGO clinic/hospital | 0.83 (0.74-0.92) | 0.001 | 0.92 (0.80-1.07) | 0.267 |
| Upazila Health Complex | 0.76 (0.64-0.92) | 0.005 | 0.82 (0.64-0.98) | 0.047 |
| District Hospital | 0.79 (0.66-0.94) | 0.008 | 0.87 (0.71-1.08) | 0.210 |
| **External sources of revenue** | | | | |
| None | 1.00 | | 1.00 | |
| Other than Govt. | 1.14 (0.91-1.43) | 0.252 | 0.94 (0.67-1.32) | 0.732 |
| Govt. | 1.14 (0.93-1.41) | 0.217 | 0.97 (0.71-1.31) | 0.829 |
| **External supervision (Past 6 months)** | | | | |
| No | 1.00 | | 1.00 | |
| Yes | 0.95 (0.87-1.04) | 0.293 | 0.94 (0.82-1.08) | 0.387 |
| **Routine quality-assurance activities** | | | | |
| No | 1.00 | | 1.00 | |
| Yes | 1.02 (0.96-1.09) | 0.493 | 1.01 (0.93-1.10) | 0.739 |
| **Client fees** | | | | |
| None | 1.00 | | 1.00 | |
| Separate fees | 1.05 (0.95-1.16) | 0.316 | 1.14 (1.01-1.28) | 0.036 |
| Fixed for all services | 1.07 (0.94-1.21) | 0.309 | 1.10 (0.91-1.33) | 0.326 |
| **24-hour provider availability** | | | | |
| No | 1.00 | | 1.00 | |
| Yes | 1.07 (0.99-1.15) | 0.087 | 1.09 (0.99-1.19) | 0.092 |
| **Client feedback system** | | | | |
| No | 1.00 | | 1.00 | |
| Yes | 1.12 (1.03-1.21) | 0.009 | 1.17 (1.04-1.30) | 0.005 |
| **Ability to provide HT care** | | | | |
| Only diagnosis | 1.00 | | --- | |
| Only treatment | 1.08 (0.95-1.24) | 0.239 | | |
| Both diagnosis and treatment | 0.98 (0.90-1.07) | 0.667 | | |
| **Integrated HT–DM service capacity** | | | | |
| Only diagnosis for both conditions | --- | -- | 1.00 | |
| Only treatment for both conditions | | | 128 (1.04-1.52) | 0.011 |
| Both diagnosis and treatment for both conditions | | | 1.10 (0.97-1.25) | 0.140 |
| Mixed/partial services | | | 1.01 (0.84-1.17) | 0.937 |
| **No. of HT providers** | 1.01 (1.00-1.02) | 0.049 | --- | |
| **No. of HT & DM care providers** | --- | --- | 1.00 (0.99-1.01) | 0.485 |

IRR = Incidence Rate Ratio; CI = Confidence Interval; ¹ Model 1: HT readiness (dependent variable = HT readiness score, 0–8); ² Model 2: Integrated readiness (dependent variable = combined HT–DM readiness score, 0–16). Both models used negative binomial regression. All estimates are adjusted for sampling weights

disparities in performance across facility types- within a context where the establishment of NCD corners was still at an early stage. These context-specific insights have not been fully discussed in earlier regionally comparative studies.

By explicitly linking HT and DM readiness into a unified analytical framework bridges an important evidence gap and provides country-specific information that can directly support the formulation of the National NCD policies, planning, and resource allocation. The works also respond to global priorities of strengthening integrated primary care for chronic diseases under the WHO PEN framework and provide a crucial baseline for monitoring Bangladesh's progress towards Universal Health Coverage.

## Comparison with other studies

The very low HT readiness score in our study should be interpreted considering the policy environment that existed at the time of the survey. The Government of Bangladesh announced an initiative in 2012 for setting up NCD corners in secondary and tertiary facilities, but the implementation was slow [30]. By December 2021, only 54 NCD corners were functional, with plans to expand to 300 nationwide [30,31]. The BHFS 2017 data thus captures a period before the full implementation of dedicated NCD infrastructure, highlighting pre-reform conditions. The survey further indicated that no structured government policy or program for NCD services had been adopted at upazila or lower-level hospitals during the survey period, leading to poor readiness in these facilities.

Our findings on low HT readiness align with broader assessments of NCD service readiness in LMICs. A study from Tanzania found primary care facilities struggling to provide essential HT care due to lack of medicines, diagnostic tools, and adherence to treatment guidelines [32]. Similarly, in India, low NCD readiness in primary healthcare facilities was attributed to weak infrastructure, staff shortages, and limited medicine availability [33]. These findings highlight the common challenges confronted by health systems in LMICs in managing the rising burden of NCDs, particularly HT.

Our research found deficiencies in HT management guidelines and trained personnel in Bangladesh, which is in sync with the wider challenges of HT care in resource-poor settings. Effective HT management depends on standard treatment protocols and trained health-care providers [9]; however, these two critical components are inadequately supported in the health system of Bangladesh, leading to a low HT readiness score.

The relatively high availability of basic diagnostic equipment- such as BP apparatus, weight scales, and stethoscopes in our study suggests that most of the health system has at least the capacity to detect HT early. This is an advantage in comparison to some settings where even these basic diagnostic tools are lacking [34]. The situation of shortage regarding anti-hypertensive medicines like ACE inhibitors, thiazide diuretics, and calcium channel blockers remain a huge barrier for effective management of HT. This barrier is further represented in LMIC studies, where poor supply chains, lack of funding, and competing priorities hinder effective NCD management [32,35–37].

Facilities that implemented client feedback systems were more likely to be ready for HT care, highlighting the growing role of patient-centered care in improving service delivery. Obtaining feedback from patients helps to identify gaps in service delivery and areas that need attention, thereby making the health system more responsive and better prepared [38]. The number of HT care providers was positively associated with readiness, suggesting that increasing staffing can address some gaps in HT care readiness. Similar findings were consistently noted by other studies on facility readiness across different disease domains [24,35,39,40].

The findings of this study have indicated that private clinics or hospitals have a higher readiness concerning HT as compared to other types of facilities. This shows that private clinics or hospitals in Bangladesh might possess comparatively much better capacity for managing HT owing to better financial resources, infrastructure, and staff training as compared to NGO-run and public facilities [41,42].

The findings suggest that, while the HT readiness score was at 46.5%, the integrated HT-DM readiness score was at 41.9%, which means that facilities are better ready for HT-only care than for integrated care. Thus, the Bangladeshi health system is not yet adequately equipped to manage both HT and DM together, reflecting systemic barriers in staff training, diagnostic tools, and availability of medicines for the management of both conditions.

 

Facilities offering treatment-only services for both HT and DM and those that reviewed client feedback were more likely to be ready for integrated care. Facilities that had service fees set separately also showed greater readiness in the integration process. These imply that, among the elements that boost integration readiness, financial mechanisms, patient engagement, and the delivery model need to be focused upon.

Private health facilities are exhibited better ready for integrated HT-DM management compared to NGO clinics. This indicates that public-private partnership initiatives might play a fundamental role in improving integrated HT-DM care within the health system of Bangladesh. However, there are major challenges in these areas, particularly with public and NGO facilities, where these issues deserve targeted interventions focusing on strengthening integrated management of coexisting chronic conditions such as HT and DM.

Although the dataset is from 2017, these findings remain relevant and informative. They capture the structural readiness of the health system before Bangladesh's major NCD initiatives, offering a foundation for tracking progress since reforms began. Moreover, persistent system gaps—such as shortages of trained staff, absence of standardized treatment protocols, and inconsistent drug supply—continue to be reported in more recent assessments, confirming that the challenges identified remain policy priorities.

## Policy and system implications

The results indicate that there remain persistent human resources and systems challenges in Bangladesh's response to NCDs, particularly HT and DM. For these insights to be acted upon, an array of feasible, context-specific strategies is required in accordance with the national priorities for health systems strengthening.

*First*, the measures to address the shortage of trained personnel should include institutionalization of in-service training and continuous professional development (CPD) for health care providers. Upon the short modular training on HT and DM management-facilitated through district and upazila health facilities with integration into the existing NCD corner-setting textbooks or in consultation with professional academic bodies such as the Bangladesh College of Physicians and Surgeons-would sustain and standardize this system.

*Second*, a national adaptation and dissemination of simplified WHO-PEN protocols would be warranted in light of the variable availability of clinical guidelines and their conflicting implementation in different facilities. The integration of such protocols across digital decision-support systems and supervisory checklists can ensure adherence in settings with scarce resources while fostering service delivery consistency.

*Third*, the absence of a number of essential antihypertensive and anti-diabetic medicines points toward continued challenges in procurement and supply chain governance. The Directorate General of Health Services (DGHS) can use centralized forecasting and electronic logistics management systems for the continuity supply chain of NCD medicine. Further pooling procurement mechanisms in the public-private and NGO sectors would lower costs and enhance the availability of medicines across the country.

*Fourth,* the positive link between customer feedback mechanisms that correlate with increased readiness emphasizes the need for community participation and accountability. Institutionalizing low-cost mechanisms—for instance, community scorecards, grievance boxes, or patient exit interviews—will help reveal gaps in services, raise responsiveness, and support public confidence in the health system.

*Fifth,* the greater readiness observed in the private facilities underlines the merits of fostering public-private partnerships (PPP). A partnership could entail joint human resource training initiatives, contracts to support diagnostic services, or collaborative procurement of medicines. These outreach indicators can be integrated into the existing accreditation frameworks of DGHS so that the private sector activity can align with our national quality standards.

*Lastly*, HT and DM care should be integrated into primary health care via multi-sectoral collaboration covering nutrition, lifestyle modification, and health promotion. The wide network of community health workers in Bangladesh could be leveraged to provide early screening, continuous follow-up, and referral linkages between households and health facilities.

All these interventions will enhance Bangladesh's role in chronic disease prevention and management through an integrated, equitable, and sustainable health system supporting Universal Health Coverage and national NCD control strategies.

## Strengths

The key strength of this study is the nationally representative BHFS 2017 data, which assures that the findings are generalizable to health facilities across Bangladesh. Adoption of a standardized WHO–SARA based readiness index will bring the strengths of external validity through direct comparison of this study with other countries and previous readiness assessments. Most relevant things about this study are: while some multi-country analyses on NCDs such as DM and cardiovascular disease have already been done using this same BHFS data, this one's focus on the readiness in HT and integrated HT–DM readiness provides quite unique insights about the country policy context, determinants, and system-level gaps that may not be easily visible in pooled or regional analyses. Besides, the integrated HT–DM readiness combine is highly analytical, which forms key base evidence for tracking future policy progress and designing strategies to strengthen Bangladesh's NCD service delivery.

## Limitations

Despite the strengths of this study, there are limitations. *First*, causal relationships between HT readiness and the determined determinants could not be established because of the cross-sectional study design. *Second,* since the first NCD corners were piloted in 2021 at 54 upazila health complexes, these improvements took place well after the BFHS 2017. Hence, these findings would be considered as baseline evidence reflecting pre-reform conditions. Importantly, the systemic gaps we identified- such as shortage of trained staff, lack of management guidelines together with limited availability of medicines-have continued to be reported in more recent studies on NCD services, further underscoring the relevance of our findings for health system strengthening. *Third,* using secondary data means that the information availability and quality is tied to the original survey design and data collection processes. For example, the survey did not provide details about the content or the extent of training that healthcare providers had received. *Fourth,* the readiness assessment focused on the availability of essential inputs at the facility level and did not directly assess the quality of care rendered or patient outcomes.

*Fifth*, while the BHFS 2017 considered a wide variety of facility types, for instance, district hospitals, upazila health complexes, maternal and child welfare centers, NGOs, and others run by private hospitals, the classifications did not consider diabetic hospitals as a separate facility category. In the sampling framework of the BHFS, diabetes clinics and hospitals were subsumed under the larger categories of private facilities, meaning that their data was not analyzed distinctly but rather through aggregation. Thus, these specialized centers were represented in the dataset but were unable to be isolated for their individual contribution toward the readiness of NCD services. Regardless, the BHFS 2017 remains the only national-level health facility readiness survey in Bangladesh, and thus, its findings would still offer some crucial baseline evidence on systemic gaps in facilities frequently accessed by the general population. *Finally,* a further limitation of this study would include the 786 facilities that had been excluded from the original BFHS 2017 sample. These facilities were excluded because they were below the sub-district level, where generally services for NCDs are not provided, or else they had missing information on key readiness indicators required for constructing the composite score. Although the exclusion reduces the overall sample size and may introduce selection bias, the remaining 382 facilities are still a nationally representative distribution of sub-district and higher-level facilities that are the primary providers of HT and DM care in Bangladesh. However, care should be taken while drawing conclusions from this study because the exclusion of lower-level facilities and those that provided incomplete data may limit how far the results can be generalized to all health facilities in the country.

## Conclusions

This nationally representative study assessed health facility readiness in Bangladesh for HT and integrated HT–DM care, revealing major gaps in trained personnel, clinical guidelines, diagnostics, and essential medicines. Although BP apparatuses and other basic equipment are widely available, overall system readiness for both single HT and integrated HT-DM care remain low. Facility type, client feedback mechanisms, and number of HT care providers contributed significantly towards HT readiness, thereby proving the importance of the capacity of the private sector, patient engagement, and strength of the workforce. For integrated HT–DM readiness, the key determinants were types of facilities, service focus (treatment-only), client feedback, and user-fee structure, which imply that organized service delivery, financial mechanisms, and accountability systems enhance readiness. The integrated readiness was considerably less than the HT readiness, indicating that health facilities are still not organized for complete management of chronic diseases. Hence, there is an important need to strengthen integrated training, dissemination of standardized guidelines, ensuring continuous supply of medicines, and institutionalizing patient feedback systems. Public–private and NGO collaborations can expedite progress in this regard. Dedicating itself to Universal Health Coverage, Bangladesh should prioritize improving readiness for integrated HT–DM care in the next phase toward equitable and sustainable management of NCDs.

## Acknowledgments

We are thankful to the DHS measures for freely granting 2017 BHFS data access.

## Author contributions

**Conceptualization:** Jannat-E-Mim Jisan, Mosiur Rahman.

**Data curation:** Jannat-E-Mim Jisan, Mosiur Rahman, Md. Sarwar Zahan.

**Formal analysis:** Jannat-E-Mim Jisan, Mosiur Rahman.

**Investigation:** Jannat-E-Mim Jisan, Mosiur Rahman, Md. Sarwar Zahan.

**Methodology:** Jannat-E-Mim Jisan, Mosiur Rahman, Prosannajid Sarkar.

**Project administration:** Mosiur Rahman.

**Resources:** Jannat-E-Mim Jisan, Mosiur Rahman, Md. Sarwar Zahan, Wazedul Islam, Md. Mamunur Rashid, Anika Tabashsum, Chowdhury Mashrur Mahdee, Fahim Sahriar.

**Software:** Jannat-E-Mim Jisan, Mosiur Rahman, Md. Sarwar Zahan, Prosannajid Sarkar, Md. Rashed Alam, Wazedul Islam, Farhana Akhter Liza, Majharul Islam, Md. Mamunur Rashid, Anika Tabashsum, Chowdhury Mashrur Mahdee, Fahim Sahriar.

**Supervision:** Mosiur Rahman.

**Validation:** Prosannajid Sarkar, Md. Rashed Alam, Wazedul Islam, Farhana Akhter Liza, Majharul Islam.

**Writing – original draft:** Jannat-E-Mim Jisan, Mosiur Rahman.

**Writing – review & editing:** Md. Sarwar Zahan, Prosannajid Sarkar, Md. Rashed Alam, Wazedul Islam, Farhana Akhter Liza, Majharul Islam, Md. Mamunur Rashid, Anika Tabashsum, Chowdhury Mashrur Mahdee, Fahim Sahriar.

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
