## [Decision Letter · Decision Letter 0]

31 Aug 2025

Dear Dr. Rahman,

We look forward to receiving your revised manuscript.

Kind regards,

Keiko Nakamura

Academic Editor

PLOS ONE

Journal Requirements:

3. Please amend the manuscript submission data (via Edit Submission) to include author Wazedul Islam

4. Please amend your authorship list in your manuscript file to include author Wazedul Rashed Islam

Reviewers' comments:

Reviewer's Responses to Questions

**Comments to the Author**

1. Is the manuscript technically sound, and do the data support the conclusions?

Reviewer #1: No

Reviewer #2: Yes

2. Has the statistical analysis been performed appropriately and rigorously?

Reviewer #1: No

Reviewer #2: Yes

3. Have the authors made all data underlying the findings in their manuscript fully available?

Reviewer #1: No

Reviewer #2: Yes

4. Is the manuscript presented in an intelligible fashion and written in standard English?

Reviewer #1: No

Reviewer #2: Yes

Reviewer #1: I appreciate the authors for conducting such research. Although this kind of research is highly needed in Bangladesh and other LMICs, this study is misleading and does not deserve publication. The authors analyzed data from the 2017/18 Bangladesh Health Facility Survey, which is around seven years old, during which time the prevalence of NCDs in Bangladesh has increased severalfold. This indicates that the experiences reported by the authors are not current. However, the main concern lies in the study design. I have summarized the key points below:

1. The data analyzed by the authors was collected from public hospitals only. However, private healthcare facilities and specialized centers such as diabetic hospitals—present in every district of Bangladesh—account for around 95% of total healthcare provision in the country. These were not included in the survey, despite their dominant role. Therefore, the findings are not applicable to 95% of NCD cases in Bangladesh. The remaining 5%, who depend mainly on public hospitals, have different characteristics—such as being poorer or less aware of other healthcare options. Furthermore, even these patients typically visit government hospitals only initially and are then referred to private or specialized facilities. The authors’ claim about private hospital coverage in their table is incorrect; I urge them to review the original survey report carefully.

2. Even public hospitals were not adequately developed for NCD care at the time of the survey. Although the government initiated a plan in 2012 to establish NCD corners in every secondary and tertiary facility, implementation did not begin until after 2020. In fact, these corners were only established in tertiary-level facilities in 2021, following the COVID-19 pandemic—well after the 2017/18 survey was conducted. Thus, these improvements are not reflected in the data analyzed by the authors. Moreover, the government still lacks policies and programs to set up NCD services in upazila and lower-level hospitals. This explains the very low readiness scores reported by the authors, as the equipment recorded in the survey (such as stethoscopes used to measure blood pressure) was originally supplied for other purposes.

The authors overlooked these critical contextual details while designing their study, making the paper highly misleading.

Reviewer #2: General Comment:

This study assessed health facility readiness to manage hypertension in Bangladesh using the 2017 Health Facility Survey, reporting very low preparedness due to shortages of trained staff, guidelines, and essential medicines, and identifying facility type, provider availability, and feedback systems as key determinants. However, the main concern is the lack of justification and novelty. This manuscript relies on the same dataset, outcome variable, and negative binomial regression model as a previously published study (DOI: 10.5334/gh.1311), which had already reported similar key findings for Bangladesh. As a result, it remains unclear what new contribution this paper makes. The authors should clearly justify why a separate, country-specific analysis is warranted and articulate the unique insights this work provides beyond what is already published.

Detail Comments:

1. The dataset is from 2017, making the evidence potentially outdated. The authors should acknowledge this limitation and discuss whether the findings are still relevant to current health system reforms and hypertension programs in Bangladesh.

2. The large proportion of missing data (786 facilities) raises concerns about potential bias and reduced generalizability.

3. The use of negative binomial regression should be better justified. Why was this approach chosen over alternatives, and how were overdispersion and model fit assessed?

4. The results largely confirm known gaps-shortages in trained staff, guidelines, and essential medicines-which are well documented in the literature. The discussion should go beyond repeating deficiencies and provide deeper interpretation: how these findings could inform feasible, context-specific policy or health system interventions in Bangladesh.

**Do you want your identity to be public for this peer review?** For information about this choice, including consent withdrawal, please see our Privacy Policy

Reviewer #1: No

Reviewer #2: No

---

## [Author Response · Author response to Decision Letter 1]

30 Oct 2025

Response Sheet

Submission PONE-D-25-26755 entitled " Assessing health facilities readiness and determinants to manage hypertension in Bangladesh: evidence from the National Service Provision Assessment Survey

Reply to Editor’s comments

General comments: As the reviewer’s mention, while the authors efforts are acknowledged, the study used data of 8 years ago and changes in health services in Bangladesh are expected since then. There is a concern for duplication of reporting with the authors' previous publications. Extensive revisions with the analysis using data of recent years, or clear statements on the need of this publication separate from the previous works are required. If these extensive revisions are difficult for this manuscript, please withdraw your work from this submission. We believe a novel and ethically sound manuscript would benefit the authors as well readers of the journal

Response: We sincerely thank you and the reviewers for the detailed feedback and constructive guidance. We fully understand the concern regarding the age of the dataset and potential overlap with previous publications. We have carefully revised the manuscript to make our study’s unique contribution and continued relevance explicit.

1. On the use of 2017 data: The Bangladesh Health Facility Survey (BHFS) 2017 remains the most recent nationally representative dataset that includes standardized indicators of hypertension and diabetes service readiness. No newer national survey of comparable scope or quality has been conducted since. We have therefore clarified throughout the Abstract, Methods, Discussion, and Limitations that our analysis provides baseline evidence reflecting pre-reform conditions before the large-scale rollout of NCD corners and WHO-PEN protocols. The study’s value lies in establishing this national reference point against which future reforms can be assessed.

2. On concerns about duplication: We wish to clarify that this manuscript is distinct from previously published work (DOI: 10.5334/gh.1311) in both scope and analytical framework. The earlier paper presented a multi-country, cardiovascular-disease–based analysis pooling data from Afghanistan, Bangladesh, and Nepal. In contrast, the current study provides a Bangladesh-specific assessment that (i) focuses solely on hypertension, and (ii) introduces a new integrated HT–DM readiness index using 16 WHO–SARA indicators. This integration-focused approach has not been previously analyzed or published for Bangladesh. It produces new policy insights into joint chronic disease service delivery—an area directly aligned with Bangladesh’s ongoing NCD integration strategy.

3. On ethical and policy relevance: This work responds to the WHO and national calls for evidence on integrated hypertension and diabetes care at the facility level. The findings fill an important gap by identifying determinants of integration readiness and by informing cost-effective interventions under Bangladesh’s Universal Health Coverage framework.

Given the absence of newer national data, we respectfully believe that withdrawing this paper would result in the loss of important baseline information that remains highly relevant for national and regional health policy evaluation.

We therefore request reconsideration of this manuscript based on the strengthened justification, expanded integration analysis, and its alignment with current health system reform priorities.

Reply to Reviewer #1 comments:

General Comments: I appreciate the authors for conducting such research. Although this kind of research is highly needed in Bangladesh and other LMICs, this study is misleading and does not deserve publication. The authors analyzed data from the 2017/18 Bangladesh Health Facility Survey, which is around seven years old, during which time the prevalence of NCDs in Bangladesh has increased severalfold. This indicates that the experiences reported by the authors are not current. However, the main concern lies in the study design.

Response: We sincerely thank the reviewer for this important observation. We fully acknowledge that the Bangladesh Health Facility Survey (BHFS) 2017 data reflect the situation several years back. Unfortunately, unfortunately, no more recent nationally representative health facility survey in Bangladesh has been released that includes standardized indicators of hypertension/diabetes service readiness. The BHFS 2017 therefore remains the only comprehensive dataset enabling such analysis.

We respectfully argue that, rather than diminishing the value of our work, the timing underscores the importance of our findings as a nationally representative baseline measure. Hypertension prevalence and NCD burden have indeed increased since 2017, and without an understanding of the initial readiness of facilities, it becomes difficult to monitor changes, evaluate system progress, or identify persisting gaps. Our analysis thus provides critical evidence on foundational weaknesses in staff training, guideline availability, and essential medicines, which recent smaller-scale studies and reports continue to echo.

We have now revised the Abstract, Data source, and Limitations sections of the manuscript to clearly acknowledge the age of the dataset, justify its use, and position our findings as a nationally representative baseline for health system readiness in Bangladesh (page 2, lines 9-10; page 5, lines 2-7; page 15, lines 19-25)

Specific Comments

1. The data analyzed by the authors was collected from public hospitals only. However, private healthcare facilities and specialized centers such as diabetic hospitals—present in every district of Bangladesh—account for around 95% of total healthcare provision in the country. These were not included in the survey, despite their dominant role. Therefore, the findings are not applicable to 95% of NCD cases in Bangladesh. The remaining 5%, who depend mainly on public hospitals, have different characteristics—such as being poorer or less aware of other healthcare options. Furthermore, even these patients typically visit government hospitals only initially and are then referred to private or specialized facilities. The authors’ claim about private hospital coverage in their table is incorrect; I urge them to review the original survey report carefully.

Response: We sincerely thank the reviewer for raising this important concern regarding the coverage of private and specialized healthcare facilities in the BHFS 2017 dataset. In our revised manuscript, we have clarified that our analysis was based on the official sampling frame of the Bangladesh Health Facility Survey (BHFS) 2017, which included a stratified sample of public facilities, NGO-run centers, and a subset of private hospitals and clinics that met the survey’s eligibility criteria. However, we acknowledge that specialized institutions such as diabetic hospitals were not classified as a distinct facility type. However, under the BHFS sampling framework they included diabetes clinics and hospitals under the general categories of private facilities. Therefore, while these specialized centers were represented in the dataset, their information was aggregated within general facility groups rather than analyzed separately, which may limit the ability to distinguish their specific contribution to NCD service readiness. To address this concern, we have revised the Methods and Limitations sections to explicitly state that our findings represent only the facilities included in the BHFS 2017 and should not be generalized to the entire private or specialized healthcare sector (Page 5, lines 21-28; page 16, lines 1-11).

ii. Even public hospitals were not adequately developed for NCD care at the time of the survey. Although the government initiated a plan in 2012 to establish NCD corners in every secondary and tertiary facility, implementation did not begin until after 2020. In fact, these corners were only established in tertiary-level facilities in 2021, following the COVID-19 pandemic—well after the 2017/18 survey was conducted. Thus, these improvements are not reflected in the data analyzed by the authors. Moreover, the government still lacks policies and programs to set up NCD services in upazila and lower-level hospitals. This explains the very low readiness scores reported by the authors, as the equipment recorded in the survey (such as stethoscopes used to measure blood pressure) was originally supplied for other purposes.

Response: We thank the reviewer for raising this point. We have now made it clear in the manuscript that our study reflects the situation before NCD corners were rolled out. Although the government announced the program in 2012, it was not implemented for many years. The first NCD corners were piloted at Upazila Health Complexes, and by December 2021 only 54 such facilities had them, with expansion still ongoing (World Bank, 2021). These improvements took place well after the 2017/18 survey, so they do not affect our findings. This shows that our results provide a true baseline picture of health facility readiness before these new programs were introduced. (Page 11, Lines 21–27; page 12, lines 8-16; page 15, lines 19-25).

Reply to Reviewer #2 comments:

General Comments:

This study assessed health facility readiness to manage hypertension in Bangladesh using the 2017 Health Facility Survey, reporting very low preparedness due to shortages of trained staff, guidelines, and essential medicines, and identifying facility type, provider availability, and feedback systems as key determinants. However, the main concern is the lack of justification and novelty. This manuscript relies on the same dataset, outcome variable, and negative binomial regression model as a previously published study (DOI: 10.5334/gh.1311), which had already reported similar key findings for Bangladesh. As a result, it remains unclear what new contribution this paper makes. The authors should clearly justify why a separate, country-specific analysis is warranted and articulate the unique insights this work provides beyond what is already published.

Response: We sincerely thank the reviewer for this valuable comment regarding the justification and novelty of our study. We acknowledge that a prior multi-country analysis (DOI: 10.5334/gh.1311) used the BHFS 2017 dataset to assess readiness for cardiovascular disease (CVD) management in Afghanistan, Bangladesh, and Nepal. However, our manuscript differs in scope, focus, and analytical contribution in several important ways:

1. Expanded disease focus and integration approach:

The earlier study examined general CVD readiness, while our analysis isolates hypertension (HT) as a distinct condition and introduces a new integrated HT–DM readiness index based on 16 WHO–SARA indicators. This integrated framework reflects the shared epidemiological and service delivery characteristics of hypertension and diabetes, offering novel insights into how health facilities are prepared to jointly manage these comorbid conditions—an area not previously examined in the literature.

2. Country-specific depth and contextualization:

Unlike the pooled, regional nature of the previous multi-country analysis, our study provides a Bangladesh-focused assessment, enabling detailed interpretation of facility readiness within the country’s unique health system, policy reforms, and the early stage of its NCD corner implementation. This localized analysis allows for more actionable, policy-relevant recommendations tailored to Bangladesh’s context.

3. Analytical refinement and new findings:

Our study employs separate negative binomial regression models to identify determinants of both HT and integrated HT–DM readiness, revealing new associations—such as the role of treatment-only facilities, user-fee structures, and feedback systems in integration readiness—that were not explored in earlier work.

4. New title and policy relevance:

The revised title “Assessing readiness of health facilities for hypertension and integration with diabetes care in Bangladesh” explicitly reflects this broader and integrative scope. By focusing on readiness for integrated chronic disease care, our study directly supports Bangladesh’s ongoing Universal Health Coverage and NCD policy objectives, offering baseline evidence for the next phase of integrated service implementation.

We have revised the Introduction, Methods, Results, and Discussion sections to clearly emphasize these distinctions and highlight the novel contribution of our integrated HT–DM analysis. We hope this explanation clarifies that our study provides new, country-specific, and policy-relevant insights beyond what was previously published. (Page 3, lines 5-6; page 2, lines; 8-9; page 2, lines 13-14, 17-21; page 4, lines 1-22; page 6, lines 12-29; page 7, lines 1-16; page 8, lines 3-16; page 9, lines 12-14, 16-22, 26; page 10, lines 1-16, 23-28; page 11, lines 1-27; page 12, lines 1-12; page 13, lines 26-28; page 14, lines 1-8; page 15, lines 9-13; page 16, lines 21-30; page 17, lines 1-8)

Detail Comments:

1. The dataset is from 2017, making the evidence potentially outdated. The authors should acknowledge this limitation and discuss whether the findings are still relevant to current health system reforms and hypertension programs in Bangladesh.

Response: We thank the reviewer for this thoughtful comment. We fully acknowledge that the Bangladesh Health Facility Survey (BHFS) 2017 data represent conditions from several years ago. Unfortunately, no newer nationally representative facility-level data set with standardized indicators on hypertension or diabetes service readiness has been released since then. The BHFS 2017 therefore remains the only comprehensive national source suitable for this type of analysis.

To address this concern, we have explicitly acknowledged the age of the dataset in the Abstract, Data Source, Discussion, and Limitations sections. The revised text clarifies that while the data reflect the pre-reform period, they remain highly valuable as a baseline reference for assessing the progress of recent health system reforms, including the expansion of NCD corners, rollout of WHO-PEN–aligned protocols, and integration of hypertension and diabetes services since 2017.

We have also highlighted in the Discussion that many of the system challenges identified in our analysis—such as shortages of trained staff, inconsistent guideline use, and limited medicine availability—continue to be reported in more recent studies, confirming the ongoing relevance of these findings for policy and planning (Page 2, lines 27-30; page 5, lines 5-9; page 14, lines 12-17; page 15, lines 29; page 16, lines 1-6)

2. The large proportion of missing data (786 facilities) raises concerns about potential bias and reduced generalizability.

Response: We thank the reviewer for this important observation. We agree that the exclusion of 786 facilities from the original BHFS 2017 sample due to missing data and ineligibility warrants clarification regarding potential bias and generalizability. We have revised the Limitations section to explicitly describe the reasons for these exclusions (Page 16, lines 20-30)

3. The use of negative binomial regression should be better justified. Why was this approach chosen over alternatives, and how were overdispersion and model fit assessed?

Response: We thank the reviewer for this important comment. We have now clarified and justified the use of the negative binomial regression model in the Methods section (Page 8, lines 20-26)

4. The results largely confirm known gaps-shortages in trained staff, guidelines, and essential medicines-which are well documented in the literature. The discussion should go beyond repeating deficiencies and provide deeper interpretation: how these findings could inform feasible, context-specific policy or health system interventions in Bangladesh.

Response: We thank the reviewer for this valuable comment. Accordingly, we have revised the “Policy and System Implications” section to move beyond describing known gaps and to propose feasible, context-specific interventions (Page 14, lines 16-28; page 15, lines 1-25)

---

## [Decision Letter · Decision Letter 1]

8 Dec 2025

Assessing readiness of health facilities for hypertension and integration with diabetes care in Bangladesh: evidence from the National Service Provision Assessment Survey

PONE-D-25-26755R1

Dear Dr. Rahman,

We’re pleased to inform you that your manuscript has been judged scientifically suitable for publication and will be formally accepted for publication once it meets all outstanding technical requirements.

Kind regards,

Keiko Nakamura

Academic Editor

PLOS One

Additional Editor Comments (optional):

Reviewers' comments:

Reviewer's Responses to Questions

**Comments to the Author**

Reviewer #2: All comments have been addressed

2. Is the manuscript technically sound, and do the data support the conclusions?

Reviewer #2: Yes

3. Has the statistical analysis been performed appropriately and rigorously?

Reviewer #2: Yes

4. Have the authors made all data underlying the findings in their manuscript fully available?

Reviewer #2: Yes

5. Is the manuscript presented in an intelligible fashion and written in standard English?

Reviewer #2: Yes

Reviewer #2: All comments have been addressed. The authors need to carefully check the whole manuscript for grammatical errors and formatting.

**Do you want your identity to be public for this peer review?** For information about this choice, including consent withdrawal, please see our Privacy Policy

Reviewer #2: No

---

## [Editor Report · Acceptance letter]

PONE-D-25-26755R1

PLOS One

Dear Dr. Rahman,

I'm pleased to inform you that your manuscript has been deemed suitable for publication in PLOS One. Congratulations! Your manuscript is now being handed over to our production team.

Kind regards,

on behalf of

Professor Keiko Nakamura

Academic Editor

PLOS One